# MPPT Control of Hydraulic Power Take-Off for Wave Energy Converter on Artificial Breakwater

**Jianan Xu \*, Yansong Yang, Yantao Hu, Tao Xu and Yong Zhan**

College of mechanical & electrical engineering, Harbin Engineering University, Harbin 150001, China;
yangyansong@hrbeu.edu.cn (Y.Y.); huyantong@hrbeu.edu.cn (Y.H.); xutao712@hrbeu.edu.cn (T.X.);
zhanyong@hrbeu.edu.cn (Y.Z.)
**\*** Correspondence: xujianan@hrbeu.edu.cn; Tel.: +86-451-8256-9750

**Abstract:** Wave energy is a renewable energy source that is green, clean and has huge reserves. In order to develop wave energy resources, an oscillating buoy Wave Energy Converter (WEC) device based on the artificial breakwater is presented in this paper. In order to effectively vent the gas in the hydraulic PTO and to improve the active control capability of the PTO system to guarantee the safety performance of the system under high sea conditions, a hydraulic PTO with an active control circuit is designed. Additionally, for the Power Take-Off (PTO) system, there is a optimal damping point under different sea conditions for PTO system, so the PTO can be controlled by the Maximum-Power-Point-Tracking (MPPT) control algorithms to improve the generated power of the system. At present, the MPPT control algorithms for wave energy are mainly used to control the load of generator. However, a fixed-load storage battery is used for the load of the generator in this paper. Additionally, an MPPT control taken at a hydraulic PTO system is executed to improve the power generated by hydraulic PTO under different sea conditions effectively in this paper. The MPPT control based on the hydraulic system is conducted by controlling the displacement of hydraulic motor to achieve the optimal damping point tracking control. The control flow of the MPPT algorithm is provided. The variable step hill-climbing method is used in MPPT control algorithm in which the big step can reduce the time of tracking and the small step can increase the accuracy of MPPT control algorithm. Due to the slow stability of the hydraulic system, a filter method for hydraulic PTO power is used. In addition, the hydraulic PTO system and MPPT control are verified to be feasible with the simulation. Additionally, MPPT control based on hydraulic variable motor is easier to carry out in practical applications than the traditional control of resistance. Finally, the simulation results demonstrate that it is an effective power control strategy for hydraulic PTO system to improve the generated power.

**Keywords:** wave energy converter; hydraulic power take-off; maximum power point tracking; hill-climbing method; variable hydraulic motor

---

## 1. Introduction

Currently, with the rapid development of human society, the growing demand for energy has put forward higher requirements. For the sustainable development of mankind, it is of great importance to develop and utilize renewable and clean energy. Wave energy is one of ocean's renewable energy with huge reserves. Approximately, it could meet the electricity requirement of most countries that have enough coastlines, if it is extensively exploited [1]. There are many forms of Wave Energy Converters (WECs), mainly in the form of absorption, pressure difference, attenuation, termination, and oscillating water column [2]. Currently, the design of WEC is quite different in the method of energy extraction, but all of them require a core of WEC, called the Power Take-Off (PTO) system, to convert the mechanical

motion of the primary wave interface into a smoothed energy output. Generally, PTO can be classified into the air turbine, hydro turbine, direct mechanical drive system, linear electrical generators and hydraulic system [3–9]. Although the air turbine and hydro turbine is highly efficient, the structure of WEC is strictly required, which make it impossible to apply to the distant sea. Linear generator requires high precision of mechanical structure. Although efficiency of linear generator is high, its cost also increases. Additionally, the hydraulic PTO can effectively extract the wave energy by converting the slow, high force motion of the WECs into the high-speed rotating motion, which also has a low cost [10]. Moreover, hydraulic PTO has good control performance to improve generated power. Prof Steven Salter [11] studied the digital displacement hydraulic PTO, established a hydraulic motor model, and studied the time domain modeling of pump-motor system, which laid the foundation for the modeling of hydraulic PTO.

However, most of the current designs for hydraulic PTO do not introduce wave models, which will lead to inaccuracies in PTO design. Additionally, further it will lead to the inaccuracy of the PTO optimal damping. So, it is necessary to introduce wave model in hydraulic PTO design. Besides, most of the current hydraulic PTO circuits are closed circuits, which results in that the gas in the hydraulic system cannot be effectively vented, resulting in poor stability of the PTO. Additionally, the parameters of the hydraulic PTO used now are mostly fixed after design. It is difficult to adapt to different sea conditions, so that the generated power by the hydraulic PTO is hard to reach the maximum power.

Currently, there are a variety of control methods for ocean wave energy, such as the declutching control, Proportion-Integration-Differentiation (PID) control, fuzzy control, complex-conjugate control, latching control, predictive control and Maximum-Power-Point-Tracking (MPPT) control [12–16]. The predictive control [14] has good control performance, but it needs big data and excellent model to predict the ocean waves. Therefore, the algorithm is difficult to implement and requires certain hardware support, which will increase the cost of the equipment.

PID control and fuzzy control are common control strategies, in which the control target is needed. However, the target parameters of PTO system are uncertain in actual sea conditions. It is difficult to give a good control target for the PTO system. Additionally, because of the particularity of the PTO system, it is difficult for the PTO system to give the transfer function like other active control systems. Therefore, PID, fuzzy and other control methods are difficult to use in maximum power capture for hydraulic PTO.

At present, complex-conjugate control is a theoretical perspective on the PTO control strategy, and it cannot be applied to the actual control system. Additionally, the latching control may cause the system output to be discontinuous, and is difficult to apply to hydraulic PTO.

The MPPT control strategy is a well-known adaptive control strategy that has achieved great success in other renewable energy industries (mainly solar and wind). For applications in marine renewable energy, there have been some studies on linear generators for MPPT. However, so far, there are few publications on the use of MPPT technology for hydraulic PTO, and the current MPPT technology is mainly used in the research of generator resistance [17–21]. Since the generator resistance is generally fixed and cannot be adjusted in practical applications, the MPPT algorithm of the variable hydraulic motor is easier to implement in an actual system than the resistance-regulated MPPT algorithm. Although Amon et al. [17] compares the results of the MPPT algorithm in random waves with different step length, the fixed step MPPT method will cause the problem of long algorithm stabilization time. Various types of WECs have been developed (Drew et al. [22]) and there has been much research to maximize the absorbed power and improve their efficiency. Xi Xiao [23] et al. propose an MPPT algorithm for the direct-drive wave energy generator by experiment and simulation, which mainly adjusts the resistive property and damping coefficient of the generator. Terry Lettenmaier [24] proposed a new MPPT algorithm through the adjustment of power generation load and applied it to the WET-NZ wave energy power generation equipment to complete the offshore test research.

At present, most of the MPPT algorithms are applied to the linear generator, not for the hydraulic PTO. Additionally, the MPPT method for the linear generator is not applicable to the hydraulic PTO system due to the long stabilization time required in the hydraulic PTO.

In this paper, an oscillating buoy wave energy converter with hydraulic PTO applied to the breakwater is mainly designed. The model of hydraulic PTO is designed by introducing the 'Linear Wave Theory' by Johannes Falnes, and the response of the hydraulic system is calculated and analyzed by simulation. After adjusting the damping in the hydraulic system, the system power generation under different damping is obtained. Finally, the system adaptive optimization is realized by designing the MPPT algorithm. This paper presents a MPPT control algorithm based on the designed hydraulic PTO applied to the breakwater, and the system adaptive optimization is realized. In addition, although the adjustment process of the hydraulic system takes a long time to stabilize, the MPPT algorithm based on the hydraulic motor has a longer time to adjust to make the power generation system stabilized at a higher power condition due to the lower frequency of the ocean waves and less variation of the wave conditions in a specific area.

Realistic seas do not have periodic waves. Usually, they are described by statistical parameters such as the wave period and significant wave height. The approach of using the optimal values obtained from the height and period of regular waves, instead of these statistical parameters, as a simplification for irregular waves, is applied in this work and it can be found in others also. This paper aims to use the MPPT algorithm to maximize the generated power of hydraulic PTO without changing the buoy used. Additionally, in order to effectively simplify the model, reduce the simulation time and improve the convergence of the simulation, the linear wave theory is used to analyze the hydrodynamics of the buoy.

## 2. Design of WEC Model

The point absorber WEC based on artificial breakwater designed in this paper is demonstrated in Figure 1. The artificial breakwater is fixed by an anchor chain, so it is assumed that it is in a fixed state.

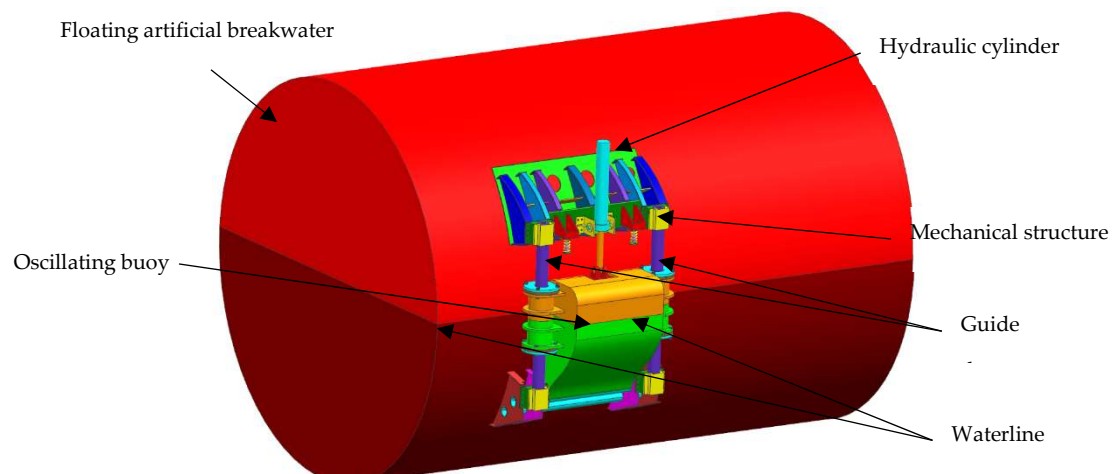

**Figure 1.** The wave energy converter model on artificial breakwater.

The WEC mainly includes three parts: oscillating buoy, mechanical structure and hydraulic PTO. The designed buoy is the first stage energy converter device of WEC. The oscillating buoy is used to convert the wave energy to kinetic energy. The main body of the mechanical structure is mainly for connecting the buoy and hydraulic system, and provides reliable support for the hydraulic PTO and the buoy. The mechanical body also connects the WEC smoothly with the breakwater, which is the basis of ensuring the stable operation of the WEC. The displacement of the buoy is limited by the guide rails on both sides, which ensures that the buoy can move up and down steadily under the action of waves to

extract the energy transmitted by waves, and then the wave energy in marine wave can be converted into the kinetic energy of the buoy through reciprocating motion. In addition, the floating breakwater is moored to the seabed, so it assumes that the buoy heaves only with a wave. The hydraulic PTO part is composed of hydraulic components such as a hydraulic cylinder and hydraulic motor. The function of a hydraulic cylinder is mainly to convert the kinetic energy transmitted by the oscillating buoy into hydraulic energy, and further convert the power through the rotary generator connected by the hydraulic motor. The main body of the hydraulic PTO is placed inside the floating artificial breakwater and connected with the hydraulic cylinder through the hydraulic pipeline.

The main working principle of WEC is that the energy absorbed by the buoy placed in the water is converted into the kinetic energy of the buoy, and the buoy pushes the hydraulic cylinder connected to it to convert the kinetic energy into hydraulic energy. The hydraulic system then converts the hydraulic energy into the rotational kinetic energy of the hydraulic motor output. Finally, the rotary generator is converted into electric energy for output.

The length of the mechanical guide is 1 m, that is, the buoy can move within ±500 mm. In order to ensure the safety of the WEC during the movement, a safety lock device is introduced into the mechanical structure to prevent the buoy from being damaged due to excessive force under high sea conditions.

## 3. Methodology

### 3.1. Mathematical Model for the Linear Wave and Linear PTO

#### 3.1.1. Mathematical Model for the Linear Wave and the Frequency Domain Analysis

The first step in designing WEC is the frequency domain analysis. Since the nonlinear theory is difficult to solve, and the wave model is not the main content of this paper, the linear wave theory is used for PTO design. Besides, in order to effectively simplify the model, reduce the simulation time and improve the convergence of the simulation, the linear wave theory is used to analyze the hydrodynamics of the buoy. Additionally, then the mathematical model for linear wave can be derived by frequency domain analysis.

According to the linear wave theory proposed by Falnes et al. [25–27], a corresponding linear wave model can be established according to the designed buoy. If the buoy is not restricted by a mechanical structure in the water, it will have six degrees of freedom, including heave, surge, sway, pitch, roll and yaw. However, the buoy reciprocates only in the direction perpendicular to the water surface due to the limitation of the mechanical structure, so only the mathematical model of the buoy dynamics under the single degree of freedom of the heave is considered.

The linear wave theory used in this paper makes a few assumptions:

Waves are two dimensional. The fluid is incompressible. There are no viscous losses. There is no underlying current. Small amplitude body motions. Wave height is much smaller than water depth or wave length.

Therefore, the designed WEC system can be analyzed by the force analysis revealed in Figure 2. The WEC model mainly includes the following forces: wave excitation force $f_e$, PTO force $f_{PTO}$, radiation force $f_r$ caused by radiation waves, and hydrostatic force $f_h$ in waves.

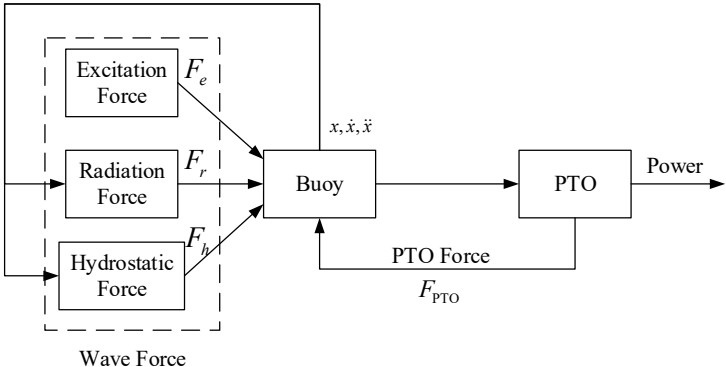

**Figure 2.** The force composition analysis of the Wave Energy Converter (WEC).

According to the force of the buoy in the wave, the force equation as shown in Equation (1) can be derived.

$$m\ddot{x} = f_e(t) + f_r(t) + f_h(t) + f_{PTO}(t) \tag{1}$$

where $m$ is the mass of the buoy. $\ddot{x}$ is the acceleration of the buoy motion. $f_e(t)$ is the excitation force of the wave input and can be calculated by the Froude–Krylov (F–K) theory. When the input waveform is an ideal sine wave, the excitation force can be represented as:

$$f_e(t) = Re(F_e e^{j\omega t}) \tag{2}$$

where $F_e$ is the amplitude of the composite excitation force and the excitation force, which is the sum of the incident wave and the diffracted wave component. Since the system is linear and has only one degree of freedom, the excitation force can be calculated by the Froude–Krylov theory with the F–K coefficient and amplitude [28].

$$|F_e| = C_{FK}\frac{H}{2} \tag{3}$$

where $H$ is the wave height, which is 2 m, applied in this paper, twice the amplitude of the wave. $C_{FK}$ is the F–K theoretical coefficient, which is related to the designed buoy shape and wave frequency. $C_{FK}$ can be defined as

$$C_{FK}(\omega) = \sqrt{\frac{2g^3\rho B(\omega)}{\omega^3}} \tag{4}$$

where $B(\omega)$ is the radiation damping coefficient, which is related to the shape and mass of the buoy and the wave frequency. $\omega$ is the frequency of the wave. The wave period of 8 s is analyzed, so $\omega = 0.785$ rad/s.

$f_r(t)$ is the radiation force of the buoy, which also is related to the shape of the buoy and the wave frequency, and can be decomposed by the same phase of the buoy acceleration and velocity as

$$f_r(t) = -A(\omega)\ddot{x} - B(\omega)\dot{x} \tag{5}$$

The coefficient $A(\omega)$ is the added mass, depending on the shape, mass of the buoy and wave frequency.

$f_h(t)$ is hydrostatic buoyancy force, which can be linearized to obtain linear hydrostatic force as

$$f_h(t) = -\rho g S\dot{x} \tag{6}$$

where $\rho$ is seawater density, which is 1.025 g/cm$^3$. $g$ is the gravitational acceleration with 9.8 m/s$^2$. $S$ is the cross-sectional area of the buoy in the direction of motion, that is, the cross-sectional area of the buoy in contact with the surface of the water, which is numbered 2.4 m$^2$.

It can be seen from the above analysis that the frequency domain analysis is an important step in WEC design and the basis of PTO model design. Therefore, hydrodynamic analysis of the buoy used was carried out by hydrodynamic analysis software ANSYS/AQWA. Figure 3 is a finite element model of designed buoy for an artificial breakwater WEC with a node number of 31,246 and unit number of 31,244. According to the gravity center position, moment of inertia and mass parameters calculated by UG software, the hydrodynamic solution analysis was carried out by AQWA. The added mass, radiation damping coefficient, Response Amplitude Operators (RAO) and the Froude–Krylov force coefficient of the buoy is obtained as demonstrated in Figure 4.

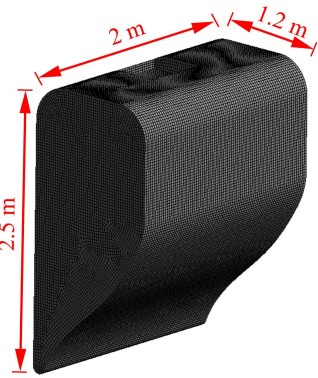

**Figure 3.** The finite element model of buoy in AQWA.

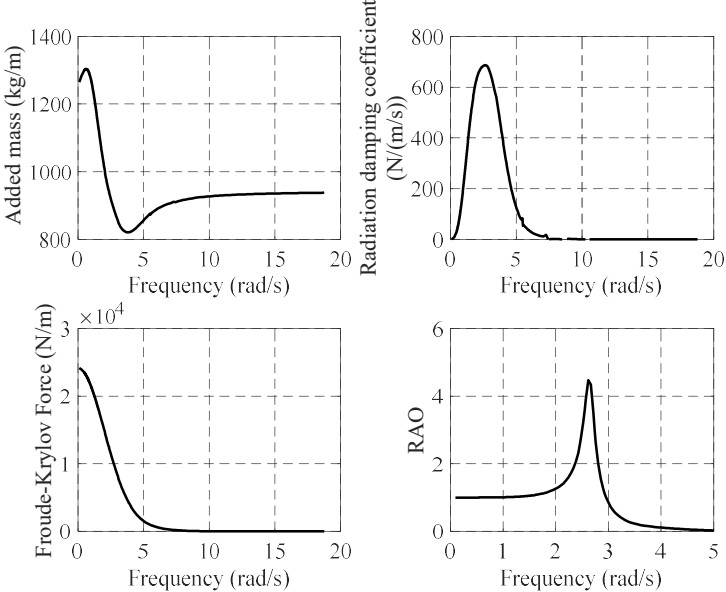

**Figure 4.** The added mass, radiation damping coefficient, RAO and the Froude–Krylov force coefficient of the buoy.

The added mass and radiation damping of the designed buoy were obtained with an 8 s wave period. The added mass of buoy was 1298 kg/m, and the radiation damping coefficient was 0.119 N/(mm/s) with an 8 s wave period. It can be seen from the analysis of RAO in Figure 4 that there was a better response for the buoy at a wave frequency of 2–3 rad/s. Additionally, the performance of PTO was studied on the basis of this buoy.

### 3.1.2. Mathematical Model for Linear PTO

When the PTO system was analyzed linearly, the PTO system was modeled as a spring-mass system. The PTO force in a linear PTO can be calculated by the spring stiffness $K$ and the damping coefficient $C$ with Equation (7).

$$f_{PTO}(t) = -Kx - C\dot{x} \tag{7}$$

The power generated with the linear PTO system is related to the equivalent damping coefficient, and the generated power of the linear PTO was calculated with Equation (8).

$$P = C\dot{x}^2 \tag{8}$$

The motion response function of the buoy can be obtained by combining Equations (1)–(3) and (5)–(7).

$$X(s) = \frac{F_e(s)}{(m+A)s^2 + (B+C)s + (\rho g S + K)} \tag{9}$$

Furthermore, the linear PTO can be analyzed by MATLAB/simulink software to obtain the response of the system with damping coefficient $C = 50$ kNs/m and spring coefficient $K = 0$, as shown in Figure 5. The displacement of buoy was 0.5 m, and the average power was 3 kW.

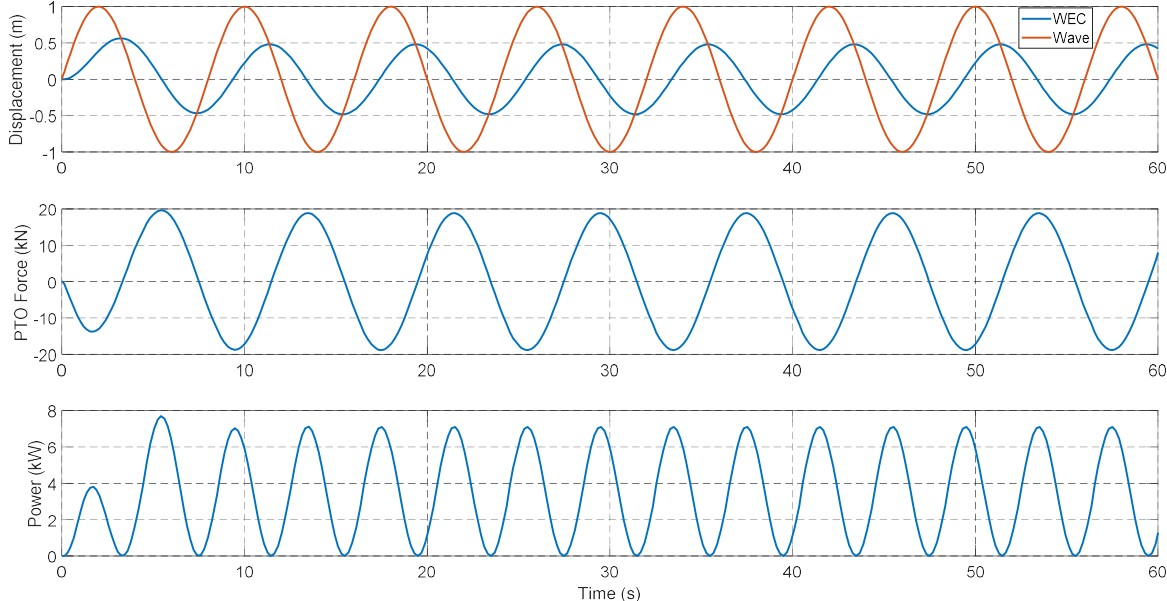

**Figure 5. Top**: Wave and WEC displacement, **Middle**: Power Take-Off (PTO) Force and **Bottom**: PTO Power for linear PTO characteristics $C = 50$ kNs/m.

### 3.2. Modelling and Analysis of Hydraulic PTO

The hydraulic PTO for WEC has the advantage of working in low-frequency high-force waves, and the power generation elements can be determined based on the power required, which is smaller, cheaper and more efficient [29]. In order to effectively vent the gas in the hydraulic PTO and improve the active control capability of the PTO system to guarantee the safety performance of the system under high sea conditions, a hydraulic PTO with an active control circuit was designed.

The schematic diagram of the hydraulic PTO system is shown in Figure 6, in which hydraulic cylinder, rectifying circuit, hydraulic motor, low-pressure accumulator, high-pressure accumulator and active control loop were mainly included.

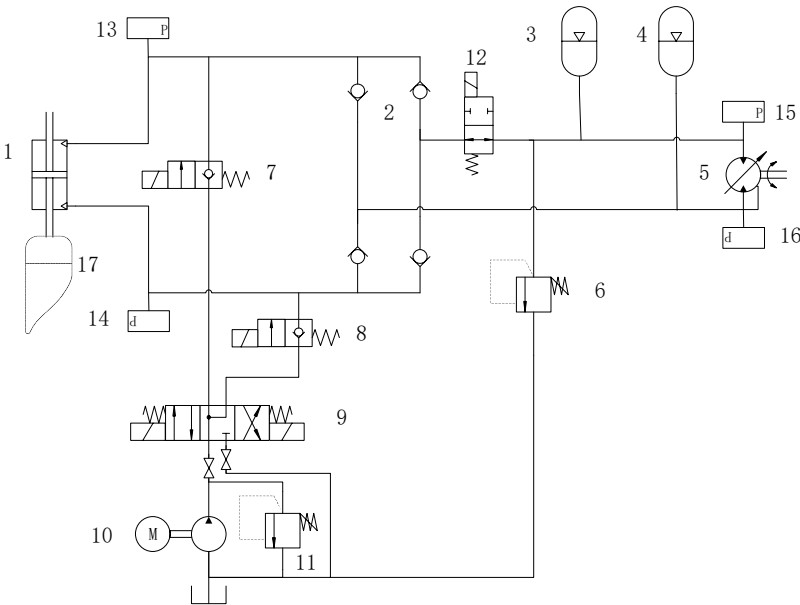

**Figure 6.** Hydraulic PTO unit circuit diagram.

In the Figure 6, 1 is a hydraulic cylinder, 2 is a rectifying circuit composed of a one-way valve, 3 is a high-pressure accumulator, 4 is a low-pressure accumulator, 5 is a variable hydraulic motor, 6 is the PTO system relief valve, 7 and 8 are electromagnetic ball valves that are inserted into check valve, 9 is an electromagnetic reversing valve, 10 is the pump station, 11 is the relief valve of pump station, 12 is an electromagnetic ball valve used in extreme sea conditions, 13–16 are the hydraulic pressure sensor and 17 is the oscillating buoy.

The hydraulic cylinder connected to the buoy converts the kinetic energy transmitted from the buoy into hydraulic energy. The rectifying circuit allows the hydraulic oil transmitted from the hydraulic cylinder to flow to the hydraulic motor in one direction, thereby ensuring the same direction of rotation of the hydraulic motor. The hydraulic motor converts hydraulic energy into rotational kinetic energy. This paper used the variable hydraulic motor, which made the motor displacement controllable. The lowest pressure of the PTO system was maintained by the low-pressure accumulator. The high-pressure accumulator was used to provide a stable pressure difference and stabilize the loop pressure to ensure the stability of the output.

The active control loop mainly includes two functions, one is to pump the oil to the PTO system when the electromagnetic valves are closed, and the other is to actively control the buoy when electromagnetic valves are opened. The active control loop mainly provides the system with a minimum pressure guarantee to prevent the system from being insufficiently supplied due to leakage, cavitation and other problems. In addition, the active control circuit controls the buoy connected to the hydraulic cylinder in extreme weather to prevent damage of the system. When encountering extreme weather conditions, opening the electromagnetic ball valves 7, 8 and 12 can actively control or lock the buoy through the 9 electromagnetic reversing valve. Additionally, because 12 is turned on, it can ensure that the accumulator and hydraulic motor will not be damaged due to excessive pressure.

Besides, the buoy can be controlled by the active control loop through the latching control method. The buoy can be locked by opening the electromagnetic ball valve 12. Additionally, the motion direction of the buoy can be controlled by the electromagnetic reversing valve 9.

According to the hydraulic system diagram, the hydraulic cylinder force, that is, the PTO force can be derived as

$$f_{PTO}(t) = (p_1 - p_2)A_P + f_v\dot{x} + f_c sign(\dot{x}) \tag{10}$$

where, $p_1$ and $p_2$ are the pressure of the piston chamber on both sides of the hydraulic cylinder. $A_p$ is the area of the piston, which is 0.0236 m$^2$. $f_v$ is the viscous friction coefficient of hydraulic cylinder. $f_c$ is the coulomb friction of hydraulic cylinder.

The captured power on the side of the cylinder connected to the buoy can be derived as

$$P_{cap}(t) = f_{PTO}\dot{x} \tag{11}$$

The generated power is defined as the power output by the hydraulic motor, which can be derived as

$$P_{gen}(t) = T_m\omega_m \tag{12}$$

where the $T_m = (p_A - p_B)D_m$ is the torque output by the hydraulic motor. The $p_A$ is the pressure of hydraulic motor inlet. $p_B$ is the pressure of hydraulic motor outlet. $D_m$ is the displacement of hydraulic motor. The $\omega_m$ is the speed output by the hydraulic motor. Both the high-pressure accumulator and the low-pressure accumulator are 20 L in volume. The pressure of the high-pressure accumulator is 3 MPa and the low-pressure accumulator is 1 MPa.

The efficiency of hydraulic PTO system can be defined as

$$\eta = \frac{\overline{P_{gen}}}{\overline{P_{cap}}} \tag{13}$$

where the $\overline{P_{gen}}$ is the average generated power, and the $\overline{P_{cap}}$ is the average captured power. There will be friction and leakage loss in the actual situation, mainly including the friction force of the hydraulic cylinder, the Coulomb friction force and the leak in hydraulic system like the hydraulic motor and the oil road. The viscous friction coefficient $f_v$ of the hydraulic cylinder was 1 kN/(m/s), the Coulomb friction $f_c$ was 2.1 kN and the inner diameter of the pipeline used was 20 mm. The wall thickness was 2.75 mm and the total length was about 10 m. The opening pressure of the check valve was 0.3 bar.

### 3.3. Design of MPPT Algorithm Based on the Hydraulic PTO

For the PTO system, there is an optimal damping point, which could be found in certain sea conditions. For the linear PTO system modeled in Section 3.1, the damping coefficient could be optimized. Additionally, for the hydraulic PTO system designed in Section 3.2, there is also an optimal damping including the motor displacement, generated damping and other hydraulic components. However, components such as hydraulic cylinders, hydraulic oil pipes and accumulators in the hydraulic system cannot be changed in the actual conditions. Not only that, but the parameters of the generator in the actual power generation system are also difficult to change.

In order to compare with the traditional MPPT algorithm, the traditional MPPT algorithm modeling method with fix step for wind energy, solar energy and linear motor is shown in Figure 7. Additionally, in order to get through the pumping oil, the initial output of the MPPT algorithm in this paper was 0.45. The traditional MPPT algorithm with fixed step did not include stability judgment and the condition of algorithm termination. The fixed step of the traditional MPPT algorithm was 0.001.

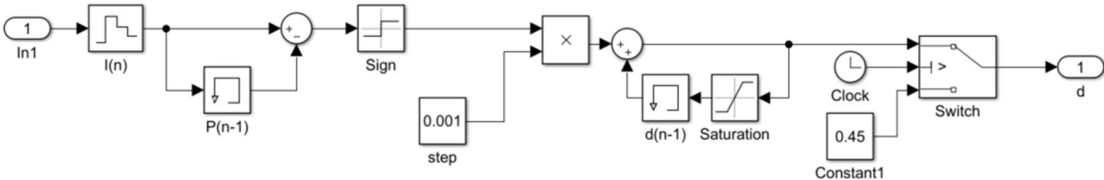

**Figure 7.** The control flow of the traditional Maximum-Power-Point-Tracking (MPPT) algorithm.

At present, the traditional MPPT algorithm has been used in solar energy, wind energy and wave energy. Many generation plant with a power converter will have an electronic MPPT system operating a very fast responding electronic PWM control with very fast ramp times, which will optimize the load on the generator. However, electronic MPPT systems are used to optimize generator load in most wave energy systems, and the MPPT control is rarely used in hydraulic PTO.

The diagram of simulation used in Simulink is shown in Figure 8. The control flow of the MPPT algorithm designed in this paper is shown in Figure 9. The variable step hill-climbing method was used in the MPPT control algorithm in which the big step can reduce the time of tracking and the small step can lift the convergence of MPPT control algorithm. Different from the traditional hill-climbing algorithm, the hill-climbing algorithm designed in this paper introduced a variable step size method, and a stable algorithm for detecting the system, further optimizing the algorithm execution rate and improving the accuracy.

The Chebyshev II and Butterworth low-pass filter were used in the designed algorithm of stability judgment. Additionally, the parameters of the filter were obtained by the spectrum analyzed through Fourier transform. Filtering not only eliminates overshoot in the system, but also provides sufficient time for algorithm analysis. Additionally, the stability algorithm was run by calculating the difference of the filtered power within a certain time Δt. The Δt was 5 s in this paper, which also could be changed to increase or decrease the stabilizing time.

In order to ignore the process of pumping oil during PTO startup, the constant output under low power was introduced into the algorithm control to ensure the algorithm was enabled after the system works stably. In addition, in the process of hill-climbing MPPT, in order to ensure that the algorithm was stable near the optimal point, the optimal setting was added, that is, the system optimal condition was set when the power was decreased or unchanged. The optimal setting was to prevent the output stability from degrading due to the long-term operation of the algorithm, so that the output power was always maintained at a high level.

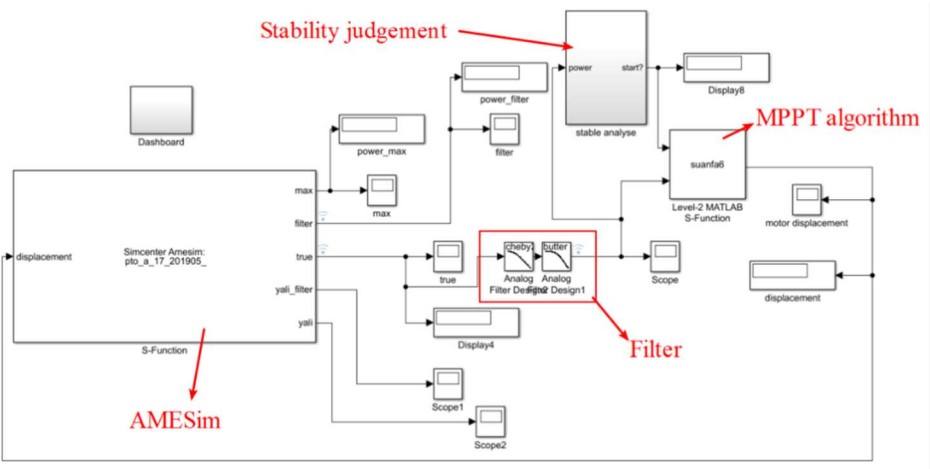

**Figure 8.** The diagram of simulation in Simulink, including modules of AMESim, stability judgment, filter and the MPPT algorithm compiled by MATLAB.

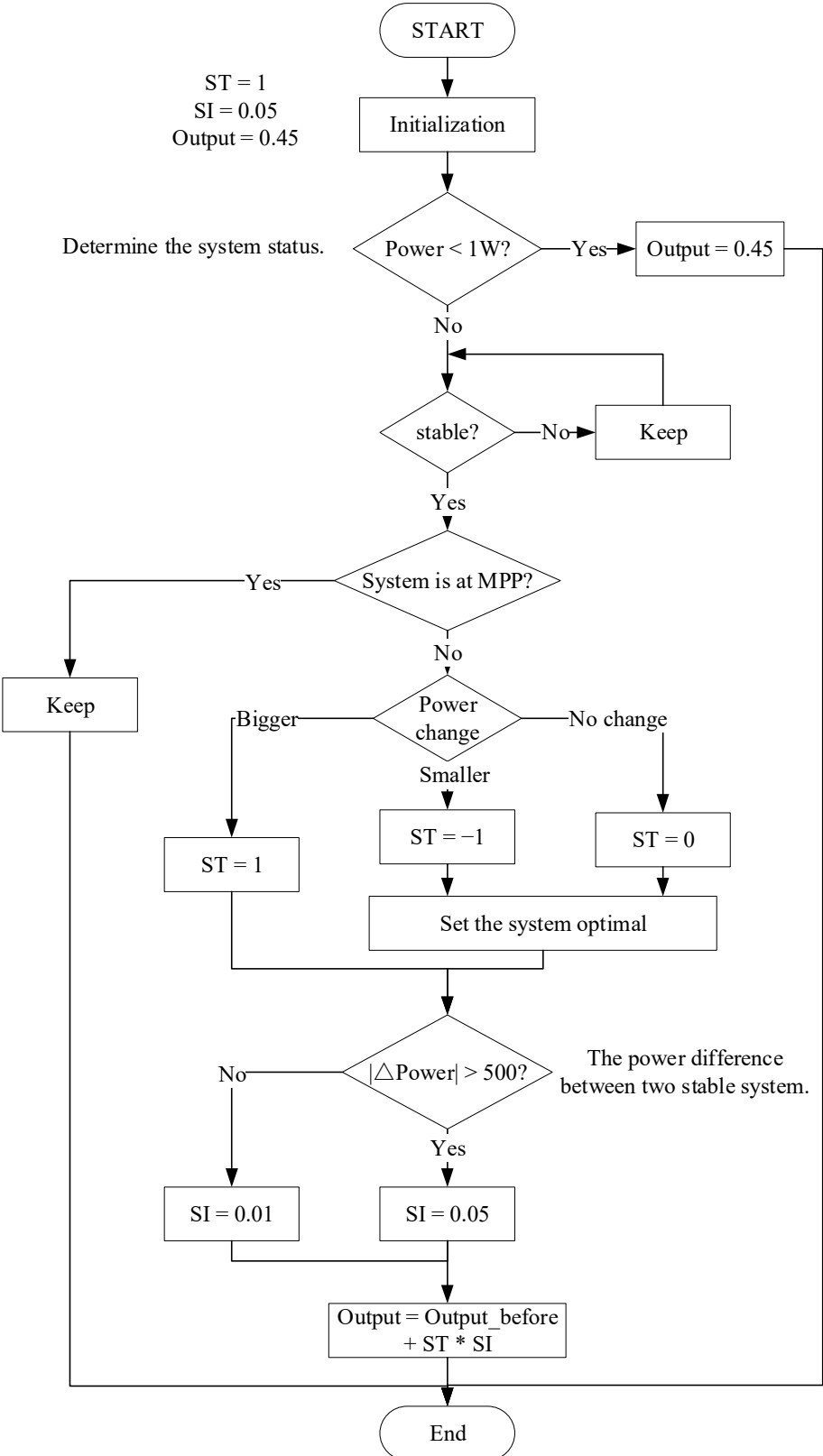

**Figure 9.** The control flow of the MPPT algorithm.

## 4. Result

### 4.1. Simulation of Hydraulic PTO

After modeling the hydraulic PTO in Section 3.2, we completed the simulation in the time domain using AMESim software. The control of the electromagnetic reversing valve and the electromagnetic ball valve was turned off in the simulation, which means the active control loop. The results including the displacement, PTO force and generated power are shown in Figures 10–13. Additionally, Figure 14 shows that the efficiency comparison of the hydraulic PTO with and without friction.

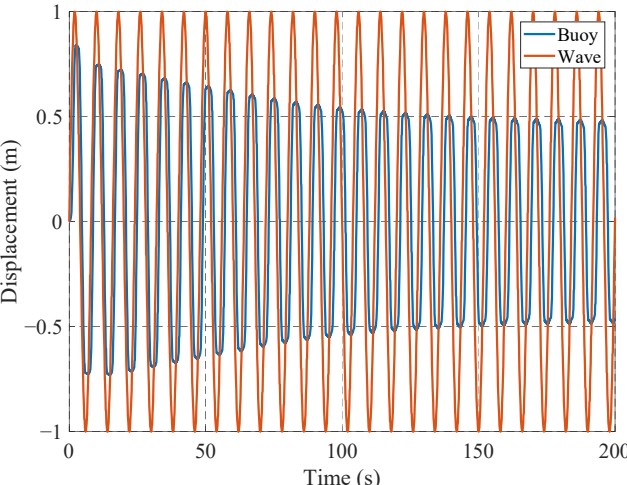

**Figure 10.** The displacement of the buoy and wave under the motor displacement is 150 cc/rev.

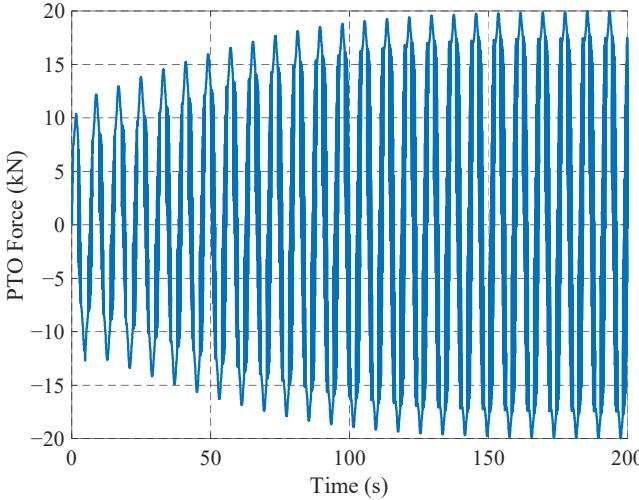

**Figure 11.** The PTO force of hydraulic PTO.

The displacement of the buoy was within ±0.5 m when the PTO system was stable. The wave height was 2 m, and the wave period was 8 s. Additionally, the displacement of the buoy gradually decreased, the main reason was that the hydraulic oil of hydraulic PTO was gradually filled up by the active circuit.

It can be seen from Figure 11 that the maximum PTO force was 20 kN when the hydraulic PTO system was stable. Additionally, there were some oscillation on the PTO force, as the wave model was introduced. However, the frequency of the oscillation was not high. In addition, the PTO force was axial, which would not cause the reliability problems.

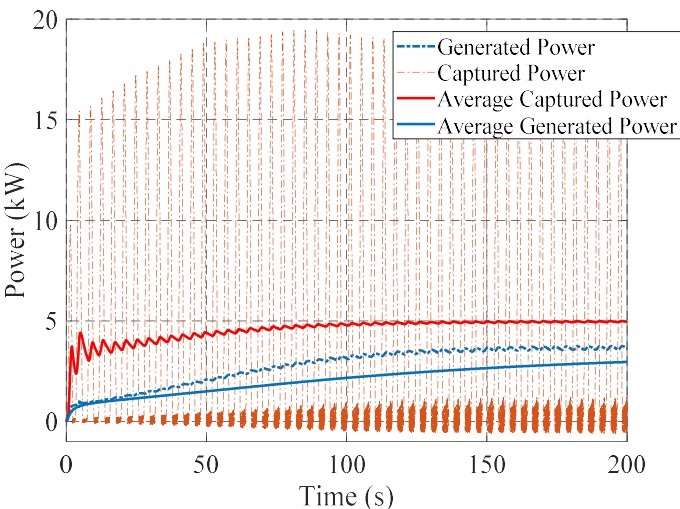

**Figure 12.** The power of hydraulic PTO. The blue line is generated power, which is the power output from the hydraulic motor. The red line is the captured power, which is the power from the buoy to hydraulic system. The generated power with friction is 3.8 kW.

It can be seen from Figure 12 that there was a certain fluctuation in the generated power of the hydraulic PTO because the accumulator could not be infinitely large. Therefore, it is necessary to use the filtering methods in the calculation of generated power.

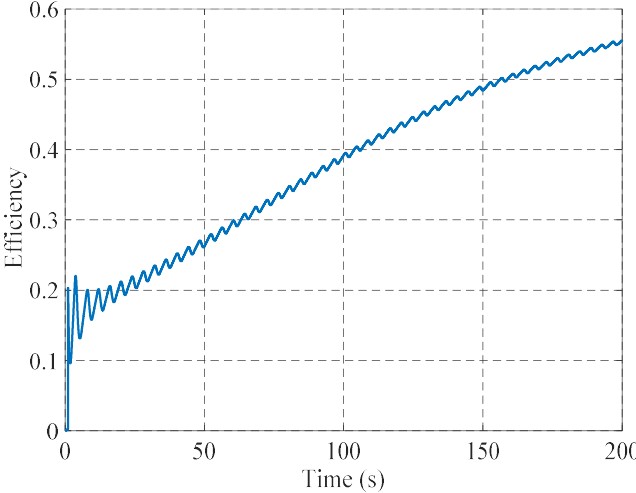

**Figure 13.** The efficiency of hydraulic PTO. The efficiency is 0.58 when the hydraulic PTO system is stable.

Due to the influence of friction mentioned above, there was a certain gap between the efficiency with and without friction as the Figure 14 demonstrated. The PTO efficiency will change near the red line by changing the parameters of the hydraulic system, but it will not exceed the blue line, which is frictionless.

All the electromagnetic valves were closed during the simulation, so the active control loop was in the charge state. From Figures 10–14, we could see that long stabilization time was required for hydraulic PTO, as the active control circuit was working for the first 150 s and the wave model was introduced.

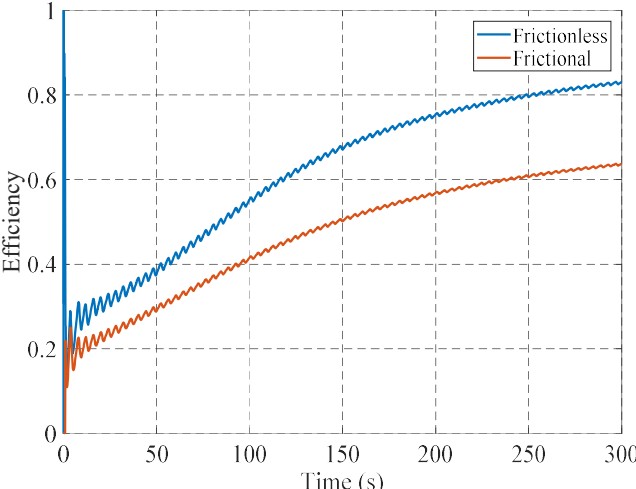

**Figure 14.** The efficiency of hydraulic PTO under frictionless and friction is 0.82 and 0.58 respectively.

*4.2. Results of Optimal Damping*

Optimal damping will be found when the damping is adjusted. We obtained the power variation of different motor displacements and generator loads through simulation.

By changing the wave height with a fixed wave period of 8 s, the optimal generated power under different wave height can be obtained as shown in Figure 15. It can be seen in the Table 1 that under the wave period 8 s, the optimal generated power point of hydraulic PTO was almost the same, that is 165 cc/rev. Therefore, the optimal damping point would not be affected by the wave height. Additionally, the maximum generated power had a linear relationship with the wave height.

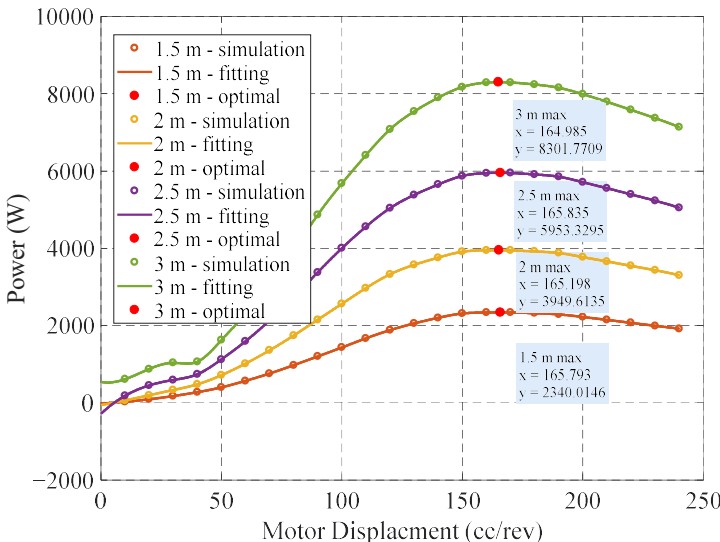

**Figure 15.** The generated power under different motor displacements and wave periods when keeping the generator load constant.

With a fixed wave height of 2 m and changing the wave period, the relationship between the generated power and the displacement shown in Figure 16 and the relationship between the generated power and the damping of the generator as shown in Figure 17. It can be seen from the two figures that the wave period had a direct impact on the optimal power point. As the wave period increased, the damping at the optimal power point also increased, showing a proportional relationship. Therefore, the optimal power point must be found through algorithms in a fixed ocean environment.

**Table 1.** The maximum generated power and its motor displacement at different wave periods.

| Wave Height | 1.5 m | 2 m | 2.5 m | 3 m |
|---|---|---|---|---|
| Generated Power | 2340 W, | 3949 W | 5953 W | 8301 W |
| Motor Displacement | 165.793 | 165.198 | 165.835 | 164.985 |

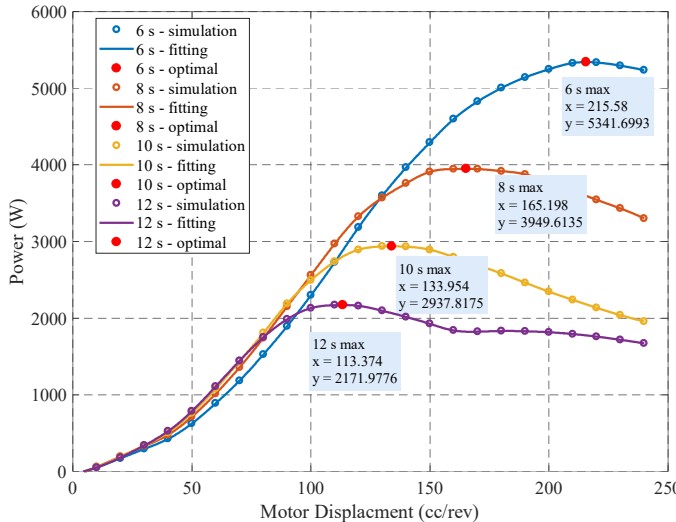

**Figure 16.** The generated power under different motor displacement and wave period when keeping the generator load constant. When the wave period is 6 s, 8 s, 10 s and 12 s, the maximum generated power is 5341 W, 3949 W, 2937 W, and 2171 W, separately.

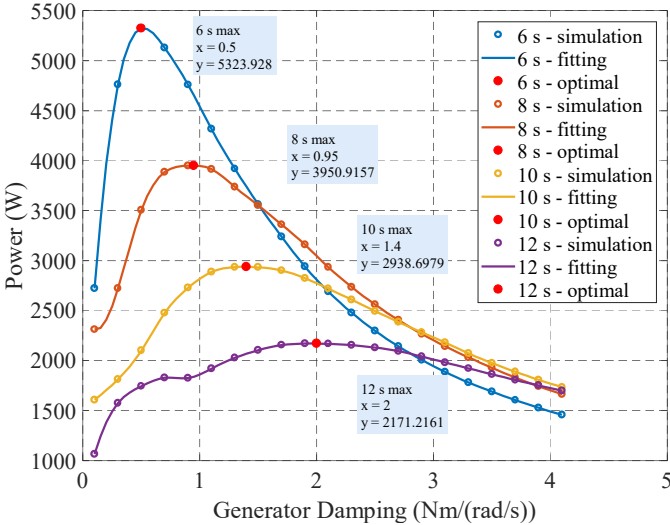

**Figure 17.** The generated power under different generator damping and wave period when keeping the motor displacement constant. When the wave period is 6 s, 8 s, 10 s and 12 s, the maximum generated power is 5323 W, 3950 W, 2938 W and 2171 W, separately.

Comparing Figures 16 and 17, we could see that the optimal power values were approximately the same under the same sea conditions, which means the maximum generated power of designed PTO was a fixed value in the fixed sea conditions. Although there were certain unavoidable deviations, they were within acceptable limits.

It can be seen from Figures 15–17 that there was only one optimal power point under each sea conditions. So, the acquisition of optimal damping lays the foundation for the MPPT algorithm. Therefore, the optimal power point in the current sea state can be tracked by the MPPT control algorithm.

### 4.3. Simulation of the MPPT Control Algorithm

The simulation of the MPPT control algorithm is carried out by the cosimulation of AMESim and MATLAB with the 8 s wave period and 2 m wave height. The result of traditional MPPT control is shown in Figure 18. The result of the designed MPPT control is shown in Figure 19.

The full displacement of the hydraulic variable motor was 300 cc/rev. Additionally, the output of the control algorithm is a ratio of the full displacement. The initial output of traditional MPPT and designed MPPT algorithm were 0.45 and perform algorithm control after reaching stability. According to the results of the traditional MPPT algorithm in Figure 18, it can be known that the optimal motor displacement obtained at the beginning was 0.5, that is 150 cc/rev, which was 15 cc/rev away from the optimal power point obtained from data fitting. Therefore, the traditional MPPT algorithm failed to get the maximum power point. Besides, after 460 s, the algorithm output gradually became 0, and the MPPT algorithm failed, because the algorithm ran continuously instead of at a stable point. Therefore, the traditional MPPT algorithm is not suitable for hydraulic PTO.

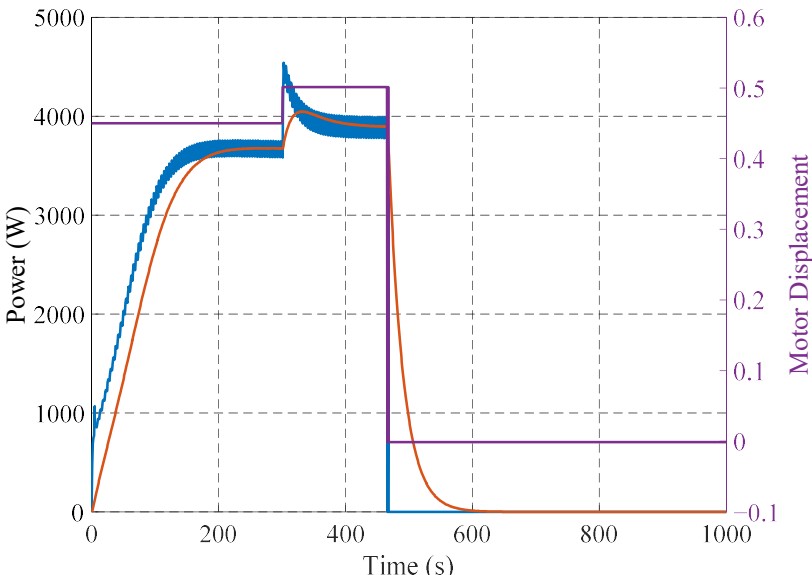

**Figure 18.** The generated power and the ratio of motor displacement under the traditional MPPT control.

However, it can be seen from Figure 18 that the optimal output obtained by the MPPT control algorithm designed in this paper was 0.54, that is, the motor displacement was 162 cc/rev. The maximum generated power achieved by the MPPT control was 3881 W. Additionally, the PTO system remained stable. The system first searched for the optimal power point through a large step, and then changed to a small step to complete the control algorithm, and the displacement was 162 cc/rev. Compared with the optimal generated power obtained by data fitting, there was only 1% difference in the optimal generated power obtained by MPPT control, which directly proves the feasibility of the algorithm.

As shown in Figure 19, the designed MPPT algorithm with stability judgment could be effectively used in the hydraulic PTO compared with the traditional MPPT algorithm.

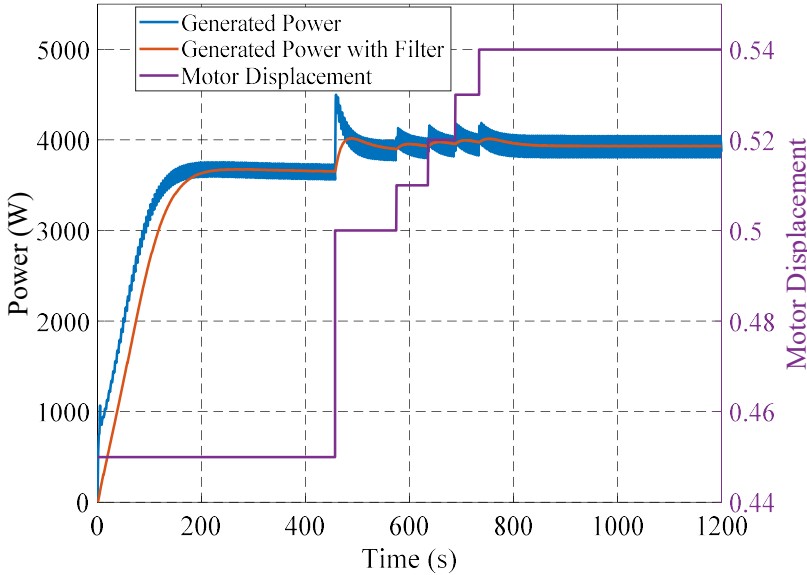

**Figure 19.** The generated power and the ratio of motor displacement under the MPPT control.

As shown in Figure 20, the hydraulic PTO efficiency under MPPT control was stable at 0.75. Compared with the efficiency in Section 4.1, the efficiency had increased by 0.17. The simulation results indicate that the MPPT control algorithm could effectively improve the efficiency of PTO and maximize the power generation.

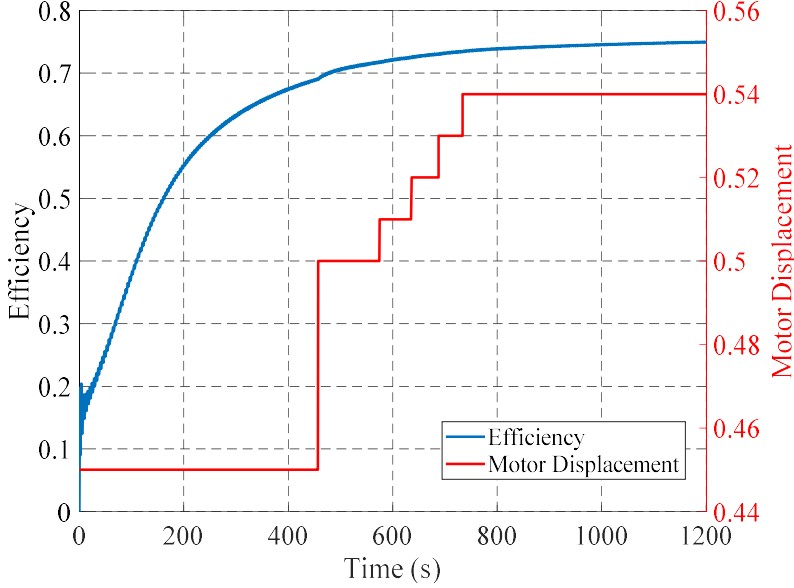

**Figure 20.** The efficiency of hydraulic PTO under the MPPT control.

## 5. Conclusions

We designed the WEC on artificial breakwater in this paper. Wave energy was captured by a buoy floating on the water, and then electricity generated from rotating generator driven by hydraulic PTO system.

Then the model of linear wave and linear PTO had been built based on the frequency domain analysis of designed buoy. Hydrodynamic parameters in the frequency domain of the buoy were analyzed by the ANSYS/AQWA software including additional mass, the radiation damping coefficient, the Froude–Krylov force and RAO. From the analysis of RAO, a better response for the buoy was

found at a wave frequency of 2–3 rad/s. The linear PTO was simulated under sea conditions with a wave period of 8 s and wave height of 2 m. The simulation results of the linear PTO show that the power was 3 kw at *C* = 50 kNs/m.

We improved the hydraulic PTO circuit by introducing the active control loops. The designed hydraulic PTO circuit could not only effectively prevent the gas in the oil circuit, but also ensured the safety of the system under extreme weather conditions. After that, the simulation was conducted by AMESim software, which verified the feasibility of the designed hydraulic circuit. The generated power of hydraulic PTO was 3.8 kW under the motor displacement was 150 cc/rev and the generator damping was 2.5 Nm/(rad/s). Additionally, the efficiency of hydraulic PTO was 0.58 when the system friction was set as described above.

Additionally, the MPPT control based on the hill-climbing method was designed to maximize the generated power. The control flow of the MPPT algorithm was provided. The filter method was used to ensure the operation of the MPPT algorithm, because there were some fluctuations in the generated power due to the inability of the accumulator to be infinite. Additionally, in the MPPT algorithm, we introduced the interrupt condition to ensure that the simulation could converge in an effective time, which also guaranteed that the output of hydraulic PTO was continued at a higher level.

The traditional MPPT algorithm was modeled and simulated by AMESim and Simulink. The results show that the traditional MPPT algorithm was not suitable for the hydraulic PTO. The results show that the traditional MPPT algorithm would fail and the optimal point was not correct, it was not suitable for the hydraulic PTO,

Finally, the feasibility of the MPPT control was verified by comparing the results of the MPPT algorithm with the fitting data of multiple simulations. The generated power with the MPPT control was 3881 W, and the hydraulic PTO efficiency was 0.75. The simulation results demonstrated that the designed MPPT algorithm could effectively maximize the generated power.

**Author Contributions:** Individual contributions include, conceptualization, J.X., Y.Y., T.X., Y.H. and Y.Z.; methodology, J.X., Y.Y., T.X., Y.H. and Y.Z.; software, T.X., Y.Y. and Y.H.; validation, T.X.; formal analysis, Y.Y., and T.X.; investigation, J.X.; resources, Y.Y., Y.H. and Y.Z.; data curation, T.X. and Y.Z.; writing—original draft preparation, Y.Y.; writing—review and editing, J.X., Y.Y. and T.X.; visualization, Y.Y.; supervision, J.X. and Y.Z.; funding acquisition, J.X. All authors have read and agreed to the published version of the manuscript.

**Funding:** This research was funded by the High Technology Ship Scientific Research Project from Ministry of Industry and Information Technology of China-Floating Security Platform Project (the second stage) grant number 201622.

**Acknowledgments:** The authors gratefully acknowledge the financial support from the High Technology Ship Scientific Research Project from Ministry of Industry and Information Technology of the People's Republic of China-Floating Security Platform Project (the second stage).

**Conflicts of Interest:** The authors declare no conflict of interest.

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
