# Peer review of "MPPT Control of Hydraulic Power Take-Off for Wave Energy Converter on Artificial Breakwater"

_jmse, doi:10.3390/jmse8050304_

Round 1
Reviewer 1 Report
I suggest to improve references related to wave energy converter:
A new solution for sea wave energy harvesting, the proposal of an ironless linear generator
Curto D. et others
Study on capture power of the sealed-buoy wave energy converter in low energy flow density area
Derong D. et others
Hybrid power generation system with Total Harmonic Distortion minimization using improved Rider Optimization Algorithm: Analysis on converters
Ravikumar, S et others
Dynamic analysis of wave action on an OWC wave energy converter under the influence of viscosity
Wang R. et others
Author Response
Thank you for your valuable comments, we have added references 5-9 and 11 that related to wave energy converter.
Reviewer 2 Report
The paper proposes a new Maximum Power-Point-Tracking (MPPT) control algorithm for the Wave Energy Converter of the floating breakwater. The algorithm gives the better efficiency of wave energy conversion. In the paper, the unit diagrams of the system are shown and demonstrated. The efficiency of hydraulic power convertor is discussed in the Setion.4. Newly developed model for hydraulic power generator is very efficient. The publication of the paper is useful to develop the offshore breakwater with a wave energy convertor. The following small question should be answered to make the discussion clear.
- Figure 1 ; Buoy oscillating is only heaving motion? The friction is assumed to be very small? What is the maximum water depth of installation of the breakwater?
- Figure 6 ; Which part is corresponding to the buoy motion?
- Figure 18 ; What is the most characteristic difference between the traditional and newly proposed MPPT control system ?
- Line 405; “Figure 18” is right?
- Line 448; correct, it >>> correct, and it
Author Response
Q1: Figure 1; Buoy oscillating is only heaving motion? The friction is assumed to be very small? What is the maximum water depth of installation of the breakwater?
A1: We are so sorry that this part was not clear in the original manuscript. The buoy is restricted by the guide rails on both sides and the breakwater is moored. Therefore, it assumes that the buoy heaves only with wave. The smooth guide rails are used to restrict movement of the buoy, so the friction is assumed to be very small. About the maximum water depth, considering this devise works in south China sea, we think it's ideal water depth and have small effect on the performance of the device. We have revised the contents of this part.
Q2: Figure 6; Which part is corresponding to the buoy motion?
A2: We are very sorry for our unclear expression, the Figure 6 has been revised in which the connected buoy is added in the revised paper.
Q3: Figure 18; What is the most characteristic difference between the traditional and newly proposed MPPT control system?
A3: As stated in lines 283-287 of the paper, the traditional MPPT algorithm does not include stability judgment and the condition of algorithm termination. The traditional MPPT algorithm compare the value of previous sampling period instead of the value when the system is stable which will cause a certain deviation in the algorithm calculation, which will cause a certain deviation in the algorithm calculation.
Q4: Line 405; “Figure 18” is right?
A4: Figure 18 is the result of the simulation under the traditional MPPT. The results show that the traditional MPPT cannot be applied to the hydraulic PTO, and the effectiveness of the MPPT designed in this paper can also be proved by comparing Figure 19.
Q5: Line 448; correct, it >>> correct, and it
A5: We are very sorry for our incorrect writing. We have made correction according to the reviewer’s comments.
Reviewer 3 Report
Abstract needs tidying up as it's abit incoherrent as it currently reads
2nd sentence in Line 10 doesn't make sence. When mentioning 'resources' the second time, which resources are you referring to?.
Line 13, what is meant by a 'stable oil pressure'
Line 18- 19 you refer to the generator having a 'fixed generator load'. In 'practical'/ real case systems, the generator load is not fixed but varies with the demand it is satisfying.
FYI. In low demand conditions, the generator load is towards open circuit conditions and in high demand conditions the load is towards short circuit conditions.
Fig.1 is difficult to interpret and understand, this should be made clearer.
MPPT is widely used in the RE sector including wave and tidal, and includes rotory generators in wave energy. Therefore line 290 needs updated. Any generation plant with a power converter will have an electronic MPPT system operating a very fast responding electronic PWM control with very fast ramp times which will optimises the load on the generator. This ensure the torque loading on the generator drive shaft is the optimum for the RE drive system powering it. You should include this in your paper.
You should also make a case why a potentially more complex hydraulic MPPT system should be used in point absorber WEC's.
When looking at hydraulic MPPT for WEC's, the authors have failed to include the significant break throughs in this topic area published by Prof Steven Salter in the 1980s - 90's, with research into Digital Displacement Hydraulic PTO's. These contributions are more appropriate for the noon-linear wave theory better representative for the operating conditions of WEC's.
Eqn 13 is only valid the volume flow rate Vcap = Vgen, therefore only when the accumulators are taken out of circuit as these will have an influence. Therefore his should be stated.
Fig 12 plots the transient and peak power captured, in order to better represent the power captured, it would be good to also include the wave period (time) averaged power capture as well. In this way, it gives a better representation of the efficiency of the system.
It would be good to see how the proposed MPPT would scale to real sea swell conditions with wave amplitudes of 2- 3m.
Author Response
Q1: Abstract needs tidying up as it's a bit incoherent as it currently reads
A1: We have carefully revised the abstract and modified it to make it more coherent.
Q2: 2nd sentence in Line 10 doesn't make sence. When mentioning 'resources' the second time, which resources are you referring to?
A2: We are very sorry for our unclear expression, and the 2nd sentence in Line 10 is briefly explain the research significance of this paper. The second ‘resources’ refers to non-renewable energy such as fossil energy, and we have revised it to made it clearer.
Q3: Line 13, what is meant by a 'stable oil pressure'
A3: "Stable oil pressure" refers to the lowest stable oil pressure of the system, which has been corrected here. And the stable oil pressure can effectively detect the system leakage and can make all the oil pipes filled with oil. We have revised it to made it clearer.
Q4: Line 18- 19 you refer to the generator having a 'fixed generator load'. In 'practical'/ real case systems, the generator load is not fixed but varies with the demand it is satisfying. FYI. In low demand conditions, the generator load is towards open circuit conditions and in high demand conditions the load is towards short circuit conditions.
A4: The generator load is not fixed in practical case systems. However, in this paper, the load of the generator in the wave energy converter is a fixed-load storage battery to store the electrical energy. And we have revised it to made it clearer.
Q5: Fig.1 is difficult to interpret and understand, this should be made clearer.
A5: We have added an explanation and text description for Figure 1.
Q6: MPPT is widely used in the RE sector including wave and tidal, and includes rotory generators in wave energy. Therefore line 290 needs updated. Any generation plant with a power converter will have an electronic MPPT system operating a very fast responding electronic PWM control with very fast ramp times which will optimises the load on the generator. This ensure the torque loading on the generator drive shaft is the optimum for the RE drive system powering it. You should include this in your paper.
A6: We have revised this part and added the comment on lines 294-297.
Q7: You should also make a case why a potentially more complex hydraulic MPPT system should be used in point absorber WEC's.
A7: We added an explanation on lines 224-226.
Q8: When looking at hydraulic MPPT for WEC's, the authors have failed to include the significant break throughs in this topic area published by Prof Steven Salter in the 1980s - 90's, with research into Digital Displacement Hydraulic PTO's. These contributions are more appropriate for the noon-linear wave theory better representative for the operating conditions of WEC's.
A8: Thanks for your comments, we have added references and introductions in the revised manuscript.
Q9: Eqn 13 is only valid the volume flow rate Vcap = Vgen, therefore only when the accumulators are taken out of circuit as these will have an influence. Therefore his should be stated.
A9: We are sorry for the error in writing this formula. The average power should be used in this formula. We have corrected the errors in the revised manuscript. And we should have explained that the Equation 13 is used to calculate the efficiency of PTO, only considering the input power and output power. The same method can also be found in other papers.
Q10: Fig 12 plots the transient and peak power captured, in order to better represent the power captured, it would be good to also include the wave period (time) averaged power capture as well. In this way, it gives a better representation of the efficiency of the system.
A10: We have modified Figure 12 in the revised paper according to your suggestions.
Q11: It would be good to see how the proposed MPPT would scale to real sea swell conditions with wave amplitudes of 2- 3m.
A11: We believe that the expectations raised are a good research direction for the next step, and we will have a further research, but currently we cannot use the models and data in the laboratory computer due to the impact of the global novel coronavirus (2019-nCoV), thank you for your valuable comments, we will continue to work hard.
Round 2
Reviewer 3 Report
The revised manuscript reads a lot better and is clearer to follow and understand.